# Mapping Pediatric Seasonal Influenza Vaccine Safety and Immunogenicity Evidence: A Systematic Review of Clinical Trials

**DOI:** 10.3390/vaccines14010032

**Published:** 2025-12-26

**Authors:** Alejandra Munoz, Briana Olivares, Yoelis Yepes-Perez, Yanping Chen, Jorge Ortiz, Maryam Amin, Mingtao Zeng

**Affiliations:** 1Department of Molecular and Translational Medicine, Texas Tech University Health Sciences Center El Paso, El Paso, TX 79905, USA; 2Francis Graduate School of Biomedical Sciences, Texas Tech University Health Sciences Center El Paso, El Paso, TX 79905, USA

**Keywords:** children, immunogenicity, influenza, safety, vaccine

## Abstract

**Background:** Influenza poses a significant health threat to children under nine, who are at high risk of severe complications. Influenza vaccination is a key prevention strategy, but pediatric trials use heterogeneous safety and immunogenicity outcomes, follow-up windows, and dosing strata that hinder meaningful cross-trial comparison. **Objective:** To map how safety and immunogenicity outcomes are defined, collected, stratified, and reported across clinical trials of seasonal influenza vaccines in healthy children aged 6 months to 8 years, and to identify reporting patterns and gaps that limit cross-trial comparability. **Methods:** Studies were identified through a structured PubMed/MEDLINE search first conducted 20 April 2025 and last conducted June 2025, following JBI and PRISMA 2020 guidelines. We included clinical trials reporting at least one safety outcome in healthy children 6 months to 8 years old. Heterogeneity in outcome definitions, follow-up windows, and dose strata precluded meta-analysis; we conducted a narrative and per-study synthesis. Risk of bias was evaluated with RoB 2 for randomized trials and ROBINS-I (V2) for non-randomized studies following Cochrane guidance. Descriptive and visual syntheses were utilized. **Results:** Of 293 records, 20 studies comprising approximately [*n* = 12,267] pediatric participants met the inclusion criteria. All included studies evaluated inactivated, egg-based seasonal influenza intramuscular vaccines. Reporting windows and dose handling varied widely. Vaccine-related serious adverse events (SAEs) were rare (only four events, with reported SAEs happening in children 6–35 months old immunized with quadrivalent formulations; all SAEs resolved and did not result in participant withdrawal from the study). No SAEs were reported in children 3–8 years old. Immunogenicity outcomes are presented as reported by each trial, with baseline and post-vaccination sampling days reproduced; no cross-trial synthesis was performed. **Conclusions:** Seasonal, inactivated intramuscular influenza vaccines show a favorable safety and immunogenicity profile in healthy children 6 months to 8 years old. However, heterogeneous outcome definitions, variable safety follow-up windows, limited dose- and priming-specific reporting, and inconsistent immunogenicity schedules substantially constrain cross-trial comparability. **Funding and Registration:** Primary funding was provided by the Eunice Kennedy Shriver National Institute of Child Health and Human Development (Grant HD109732). This review was registered in PROSPERO (registration number: CRD420251237499).

## 1. Introduction

Influenza continues to be a major global health challenge due to its high infectivity rate and its potential to cause serious illness. Infants and young children face a higher risk of complications from influenza illness, such as dehydration, pneumonia, and encephalopathy, that can prove fatal [1,2,3]. Due to the elevated influenza morbidity and mortality rates in this group, the World Health Organization (WHO) classifies children under five as a priority group for influenza vaccination [4]. Despite vaccination recommendations, influenza remains a significant burden in children worldwide. From 2010–2016, in the United States alone, most pediatric deaths occurred in unvaccinated healthy children [5,6], underscoring the importance of timely vaccination.

Several seasonal influenza vaccines are currently licensed for pediatric use, and additional candidates remain in development. These products differ in the route of administration, formulation, and valency. Products also vary by strain composition across seasons and by age-specific dosing schedules. These design choices can shape the safety profiles of the vaccines.

Influenza vaccines are typically classified by the following features: route of administration, production platform, valency, antigenic composition, and vaccine classification. Methods of administration for influenza vaccines in children include intramuscular and intranasal routes. Types of production platforms include cell-based vaccines utilizing cell lines and chicken egg-based production methods [4]. Formulations by valency refers to the number of strains included in the vaccine [4]. Different antigenic compositions are updated by season and hemisphere to better reflect circulating strains, which help address seasonal variation [7]. Lastly, influenza vaccines can be classified as either inactivated or live attenuated vaccines. Inactivated vaccines contain viruses that are no longer able to replicate due to mechanisms of inactivation during vaccine production or design [4]. Attenuated vaccines contain live but weakened viruses that replicate to a limited or localized extent in the body [4].

A clear understanding of these categories helps us understand the specific safety profiles of the available vaccine options in pediatric populations. Along with those, the maturity of the immune system and the vaccine dosing schedule can influence how well a child’s immune system mounts a response against influenza. It is recommended that children under nine years of age receive a two-dose regimen if previously unvaccinated for influenza [8]. Priming status of children under the age of nine is determined based on lifetime dosages received prior to the current season [8].

It is also important to understand how clinical trials assess safety using standardized windows of events and reporting conventions. Immediate reactions are those observed on site after each vaccine administration until thirty minutes post-vaccination [9]. Adverse events (AEs) are medical occurrences associated with the vaccine, whether or not considered vaccine-related [9]. Serious adverse events (SAEs) are those that investigators or study personnel determine to be serious based on their determination that it may jeopardize the participant [9].

Several clinical trials and systematic reviews have assessed various influenza vaccine safety outcomes in pediatric populations. Prior investigations have evaluated inactivated vaccines [10], compared safety differences between trivalent and quadrivalent versions of the vaccine [11], and examined safety profiles of vaccines produced using cell-based and egg-based vaccines [12]. While these reviews provide valuable insights, they do not examine how safety varies with different dosing schedules or across pediatric age ranges, or chart heterogeneity. The evidence from these clinical trials remains fragmented and inconsistently reported.

To fill this gap, we conducted a descriptive systematic review that maps clinical trial evidence on seasonal inactivated influenza vaccine safety and immunogenicity in healthy children 6 months to 8 years of age. This approach is intended to clarify existing evidence that already supports confident conclusions about pediatric influenza vaccine safety and immunogenicity, and where heterogeneity in trial design and reporting still limits cross-trial comparability.

## 2. Methods

### 2.1. Eligibility Criteria

Population: This review included studies involving pediatric populations aged 6 months to 8 years. Studies were eligible if they reported safety data specific to children or stratified results by age, enabling pediatric data extraction. Studies exclusively involving adults or failing to report pediatric-specific outcomes were excluded. Mixed-population studies were excluded if pediatric data could not be isolated. Only studies on healthy children were included due to the limited sample size of high-risk populations.

Interventions: Seasonal vaccines containing antigens from influenza subtype A(H1N1), A(H3N2), and B lineages, including any platforms and formulations such as (split-virion/subunit; egg-based or cell-based). Double-dose vaccination regimens or studies that reported single- and double-dose regimen safety outcomes stratified by dosages were included.

Comparators: We included studies with placebo, active, or approved comparators, and dose/valency variations within the same vaccine platform. Single-arm interventional studies were included.

Outcomes: Primary outcomes of interest were reported AEs for local and systemic reactions post-dose; unsolicited AEs post-dose; vaccine-related SAEs through six months post-final vaccination. Local specific outcomes included pain, redness, and swelling. Systemic specific outcomes included fever, cough, headache, nausea, and malaise.

Study designs/details: Only clinical trial data were included (e.g., randomized control trials, non-randomized controlled trials, single-arm interventional studies). Observational studies, reviews, commentaries, and conference abstracts were excluded. Any included studies had to report original clinical trial data. Gray literature was excluded due to feasibility constraints. Studies from any geographic region or healthcare setting were eligible, regardless of income classification or infrastructure. Hospital-based, outpatient, community, and research settings were all included.

Additional criteria: Only English-language studies were included. The search covered publications from PubMed/MEDLINE starting 31 December 2009 through 24 June 2025.

### 2.2. Rationale for Eligibility Criteria

Choices for inclusion and exclusion criteria were designed to maximize comparability across the largest feasible set of studies while minimizing the risk of bias.

### 2.3. Search Strategy and Information Sources

This review followed the Joanna Briggs Institute (JBI) Manual for Evidence Synthesis using a three-step search strategy.

The initial limited search was conducted in PubMed/MEDLINE starting 20 April 2025, through 24 June 2025. Keywords and MeSH terms included “influenza vaccine,” “children,” “safety,” “immunogenicity,” and synonyms.The PubMed search strategy was as follows: (“influenza vaccines” [MeSH Terms] OR “influenza vaccine” [All Fields]) AND (“child” [MeSH Terms] OR “children” [All Fields] OR “pediatric” [All Fields] OR “infant” [MeSH Terms]) AND (“safety” [All Fields] OR “adverse events” [All Fields]) AND (“immunogenicity” [All Fields] OR “immune response” [All Fields]).Reference List Screening: Backward reference searching was performed for all included full-text studies. References were deduplicated in EndNote and then managed in Excel for screening.

### 2.4. Screening Stages and Selection Process

Pilot Testing: A pilot test of 6 titles/abstracts was used to refine eligibility criteria.Title Screening: Two independent reviewers (AM and BO) screened all records, and discrepancies were settled by agreement.Abstract Screening: Two independent reviewers (AM and BO) screened all records, and discrepancies were settled by agreement.Full-Text Screening: Full texts of potentially relevant studies were reviewed independently by the same reviewers. Disagreements were resolved by discussion.

### 2.5. Data Collection Process

Following JBI guidance, data were charted using a structured form (Appendix A) adapted from JBI and piloted on six studies. For multi-arm trials with shared comparators, we applied pre-specified unit of analysis rules to avoid double-counting.

### 2.6. Study Risk of Bias Assessment

We used the Cochrane risk of bias assessment tool (Appendix A) for randomized trials and the risk of bias in non-randomized studies of interventions (ROBINS-I V2) (Appendix A) for open-label or single-arm trials.

### 2.7. Data Synthesis

We pre-specified a narrative synthesis. For outcome reporting, we did not pool results and did not do quantitative data synthesis. The data is presented as the percentage of participants (%) who experienced the event in that category with, in parentheses, participants who experienced one event (n) over the total participants (N) in that study. For figure-only reports of outcomes, two reviewers (AM, BO) independently estimated arm-level percentages from published graphs by visual interpolation to axis tick marks. Recorded values were rounded to the nearest whole percent. When the two estimates from each reviewer differed by more than two percentage points, disagreements were handled by consensus. Planned subgroups were age groups of 6–35 months and 3–8 years and priming status of children, such as influenza vaccine-naïve and primed. Sensitivity analysis excluded high-risk-of-bias trials. All data extraction was conducted using Microsoft Excel.

### 2.8. Reporting Bias Assessment, Effect Measures, and Certainty Assessments

We summarized study-level safety and immunogenicity outcomes as counts and percentages. We did not pool any effects due to heterogeneous interventions, windows, and reporting formats. Because no meta-analysis was performed, we did not assess reporting bias, effect measures, or certainty assessments.

## 3. Results

The initial database search yielded 293 records. A three-stage screening process was conducted to determine eligibility (Figure 1). During the first phase, 115 studies were excluded based on their titles, primarily because they did not pertain to influenza vaccination in pediatric populations. In the second phase, 33 studies were not retrieved due to no English translations being available or due to not pertaining to our intervention and outcomes of interest. In the final full-text screening phase, 37 studies were excluded due to the dates of the recruitment process being before 2010, and 28 studies were excluded due to having a different population of interest, such as children with chronic illness. A total of 28 studies were excluded since they had a different intervention of interest, such as H5N1 or coadministration of different vaccines, and 18 studies were excluded since they either used only a single dose regimen or reported pooled single- and double-dose safety outcomes. Some 14 more studies were excluded due to having age ranges not corresponding to our inclusion criteria or not reporting stratified age outcomes that could be extracted for the population of interest. This process resulted in 20 studies meeting the inclusion criteria and being retained for data charting and synthesis.

### 3.1. Study Characteristics

The analysis included a total of 20 studies, with studies spanning only one influenza season. The majority of studies were randomized control trials, with randomization reported in most cases. However, blinding was inconsistently applied, with some studies being open-label, observer-blinded, or double-blinded. A couple of studies stratified patients by prior vaccination history, study enrollment site, or dosages. Table 1 and Table 2 show the summarized main characteristics of the included studies.

Study sites were primarily conducted in China (*n* = 6) [14,15,16,17,18,19], followed by India (*n* = 3) [20,21,22]. Most studies only had one country of recruitment (Table 1), with one study recruiting in three countries [23] and two studies including seven different countries in their trial [24,25]. Following the geographic distribution of studies, we examined the sample sizes across the trials. No studies enrolled over five thousand participants; only six studies enrolled more than five hundred participants [14,15,17,19,23,26]; and three studies enrolled less than a hundred participants [20,27,28]. Two studies in children aged 6–35 months old enrolled fewer than one hundred participants [20,28].

### 3.2. Participant Characteristics

Studies were then classified by the specific age groups of participants included. If a study covered multiple age ranges and reported stratified data for each age range, the study was included in all the age classifications it fell into. Four studies reported stratified outcomes for children only 3–8 years old [14,18,19,29] (Table 1). Five studies reported stratified outcomes for both children 6–35 months old and 3–8 years old [20,21,27,30,31]. All other studies reported stratified outcomes only for children aged 6–35 months old. In studies that reported sex distributions among populations aged 6–35 months old, 47.82% were female, and 52.17% were male. In studies that reported sex distributions for populations aged 3–8 years old, 48.53% were female, and 51.47% were male. Studies that did not report age-stratified sex distributions were not included in these sex distribution calculations.

In terms of studies conducted in children only 6–35 months old, 11 studies reported outcomes for solely influenza vaccine naïve children (Table 1), and 5 studies reported pooled outcomes for both naïve and primed populations [17,23,25,26,31]. Studies conducted in children only 3–8 years old, four studies reported outcomes for naïve populations independently [14,20,21,30]. No studies reported stratified results for primed-only populations. One study reported pooled naïve and primed outcomes for children 3–8 years old and reported naïve only outcomes for children 6–35 months old [27].

**Table 1 vaccines-14-00032-t001:** Characteristics of included clinical trials and study populations (*N* = 20 studies).

Reference	Clinical Trial Phase	Country (s)	Total Participants (*n* = 12,190)	Age Bands	Priming Status
Agarkhedkar, 2019 [20]	IV	India	100	6–35 MO	Naïve
			100	3–8 YO	
Carmona Martinez, 2014 [28]	I	Spain	20	6–35 MO	Naïve
Chang, 2020 [29]	III	Taiwan	111	3–8 YO	Naïve; primed
Chen, 2023 [14]	III	China	540	3–8 YO	Naïve
Claeys, 2018 [25]	III	Bangladesh, Czech Republic, France, Germany, Poland, Spain, US	849	6–35 MO	Naïve; primed
Cruz-Valdez, 2018 [26]	III	Mexico	134	6–35 MO	Naïve; primed
Dhamayanti, 2020 [30]	IV	Indonesia	135	6–35 MO	Naïve
			135	3–8 YO	
Diallo, 2018 [32]	II	Senegal	211	6–35 MO	Naïve
Halasa, 2015 [33]	I	US	205	6–35 MO	Naïve
Hu, 2020 [15]	III	China	2320	6–35 MO	Naïve; Primed
Kothari, 2024 [22]	III	India	346	6–35 MO	Naïve
Langley, 2015 [23]	III	Honduras, Dominican Republic, Canada	601	6–35 MO	Naïve; Primed
Mo, 2017 [16]	IV	China	150	6–35 MO	Naïve
Ojeda, 2020 [27]	III	Mexico	121	6–35 MO	Naïve
			59	3–8 YO	
Pepin, 2019 [24]	III	Czech, Dominican Republic, Greece, Honduras, Philippines, South Africa, Romania	2721	6–35 MO	Naïve
Sarkar, 2021 [21]	III	India	103	6–35 MO	Naïve
			100	3–8 YO	
Soedjatmiko, 2018 [31]	II	Indonesia	135	6–35 MO	Naïve; Primed
			135	3–8 YO	
Wang, 2024 [17]	III	China	1980	6–35 MO	Naïve; Primed
Wen, 2025 [18]	IV	China	240	3–8 YO	Naïve
Zhang, 2022 [19]	III	China	800	3–8 YO	Naïve; Primed

This table summarizes the design, geographical location, and key demographic characteristics of the pediatric populations included across all randomized controlled trials (RCTs). The total number of participants analyzed across all studies is N = 12,267. Key definitions: clinical trial phase indicates the stage of vaccine development at the time of the trial (Phase I, II, III, or IV). Age bands refers to the primary pediatric age ranges evaluated in the intervention. Further, 6–35 MO refers to children aged 6 to 35 months, and 3–8 YO refers to children aged 3 to 8 years. Priming status describes the influenza vaccination history of the cohort. Naïve indicates that participants had never received a seasonal influenza vaccine prior to the study. Primed indicates that participants had a history of prior vaccination. Abbreviations: MO, months old; YO, years old.

### 3.3. Intervention Characteristics

All 20 studies used only inactivated vaccines developed using chicken eggs for their clinical trial. Additionally, all studies only used intramuscular administration for all of their enrollees. A total of six studies used only subunit vaccines for their trial [15,20,23,25,30,31], nine studies used only split-virion vaccines for their trial (Table 2), and five studies used both types of vaccines [19,21,26,27,32]. One study evaluated an approved method for split-virion vaccine formulations against a new experimental process [25].

Studies were only included if they utilized seasonal vaccine formulations; therefore, this review only included studies that used trivalent or quadrivalent influenza vaccines (Table 2). Five studies in children 6–35 months old tested the trivalent formulation [16,26,28,32,33]. No studies in children 3–8 years old evaluated trivalent formulations. Three studies in children 6–35 months old evaluated quadrivalent formulations only [22,24,25]. One study used a trivalent formulation and reported stratified outcomes for both age bands of interest [31]. Two studies tested two different formulations of trivalent vaccines within each study (two subtypes of influenza A and one B lineage, either Victoria or Yamagata) and reported stratified outcomes for both formulations [15,17].

In our included studies, only three studies used vaccine formulations with adjuvants. One study in children 6–35 months old used AS03, of which one study tested different amounts of AS03 in each vaccine [28]. Enrollees in this study received vaccines with either 1.48 or 2.97 milligrams of AS03 reported stratified outcomes [28]. Two studies in children 6–35 months old evaluated an oil-in-water emulsion formulation (MF59) [26,32]. Two studies evaluated vaccine formulations that utilized thiomersal as a preservative for multi-dose vials in children 6 months to 8 years of age [27,31].

Another source of variation between studies is the dosages tested for the corresponding reported outcomes. Dosages used in these studies included either 7.5 or 15 µg of hemagglutinin (HA) content per strain, equating to 22.5 or 45 total µg for trivalent formulations and 30 or 60 total µg for quadrivalent formulations.

**Table 2 vaccines-14-00032-t002:** Specific features of seasonal influenza vaccine formulations (*N* = 20 studies).

Reference	Vaccine Type	Valency	B-Linage Included	HA Content (µg) per Strain	Adjuvant or Preservative
Agarkhedkar, 2019 [20]	Subunit	Quadrivalent	Both	7.5	None
				15	
Carmona Martinez, 2014 [28]	Split-virion	Trivalent	B/Victoria	7.5	AS03
Chang, 2020 [29]	Split-virion	Quadrivalent	Both	15	None
Chen, 2023 [14]	Split-virion	Quadrivalent	Both	15	None
Claeys, 2018 [25] †	Split-virion	Quadrivalent	Both	15	None
Cruz-Valdez, 2018 [26]	Subunit	Trivalent	B/Yamagata	15	MF59
	Split-virion				None
Dhamayanti, 2020 [30]	Subunit	Quadrivalent	Both	15	None
Diallo, 2018 [32]	Split-virion	Trivalent	B/Yamagata	15	None
	Subunit				MF59
Halasa, 2015 [33]	Split-virion	Trivalent	B/Victoria	7.5	None
				15	
Hu, 2020 [15]	Subunit	Quadrivalent	Both	7.5	None
		Trivalent	B/Victoria		
		Trivalent	B/Yamagata		
Kothari, 2024 [22] ‡	Split-virion	Quadrivalent	Both	15	None
Langley, 2015 [23]	Subunit	Quadrivalent	Both	15	None
Mo, 2017 [16]	Split-virion	Trivalent	B/Yamagata	7.5	None
Ojeda, 2020 [27]	Split-virion	Quadrivalent	Both	15	Thiomersal
					None
Pepin, 2019 [24]	Split-virion	Quadrivalent	Both	15	None
Sarkar, 2021 [21]	Split-virion	Quadrivalent	Both	15	None
		Trivalent	B/Victoria		
Soedjatmiko, 2018 [31]	Subunit	Trivalent	B/Yamagata	15	Thiomersal
Wang, 2024 [17]	Split-virion	Quadrivalent	Both	7.5	None
				15	
		Trivalent	B/Victoria	7.5	
			B/Yamagata	7.5	
Wen, 2025 [18]	Split-virion	Quadrivalent	Both	15	None
Zhang, 2022 [19]	Subunit	Quadrivalent	Both	15	None
	Split-virion				

This table details the technical specifications of the vaccine interventions used in the included studies. The interventions are characterized by antigen type, valency, the specific influenza B lineage included, the HA content per strain, and the presence of adjuvants or preservatives. Key definitions: vaccine type refers to the antigen structure. Split-virion means vaccines are inactivated and fragmented; subunit vaccines contain only the surface proteins (HA/NA). Valency indicates the number of influenza strains included. Trivalent (3V) includes two A strains and one B strain. Quadrivalent (4V) includes two A strains and both B lineages (B/Victoria and B/Yamagata). B-Linage includes (B/Victoria or B/Yamagata) and specifies the single B strain included in the trivalent (3V) formulation. Both indicate the inclusion of both B lineages in a quadrivalent (4V) vaccine. HA content (µg per strain): the microgram dosage of Hemagglutinin antigen contained for each influenza strain in the vaccine formulation. Abbreviations: HA, Hemagglutinin; µg, microgram; AS03/MF59, specific oil-in-water adjuvants used to enhance the immune response. None indicates no adjuvant or preservative was added. † One study compared a traditional split-virion vaccine to an investigational split-virion vaccine in which changes in the manufacturing process were implemented but did not alter the condition of virus replication or inactivation [25]. ‡ One trial compared two egg-based, split-virion quadrivalent inactivated vaccines (15 µg HA per strain), which were identical in formulation and dose and differed only by manufacturer [22].

### 3.4. Safety Outcome Data Collection

All studies for both age cohorts of interest solicited safety outcomes for a minimum of seven days after each vaccine administration. For studies in children 6–35 months old, ten administered the second dose 28 days after the first dose, and one administered the second dose on day 21 (Figure 2) [28]. One study in children 6–35 months old solicited safety outcomes for 10 days after each vaccination [33].

For studies in children only 3–8 years old (Figure 2), all collected solicited outcomes were collected for seven days after each dose. One study collected unsolicited outcomes for 180 days after the administration of the first dose [19]. All studies that reported outcomes only for this age cohort collected SAEs for a total of 180 days after the administration of the first dose. Additionally, these studies administered the second dose 28 days after the first dose. Studies varied in the follow-up period for collecting unsolicited AEs.

In studies that reported outcomes for children aged 6 months to 8 years, the second dose was administered on day 28 post-first dose in all studies [20,21,27,30,31]. One study, as shown in Figure 2, collected SAEs outcomes for 70 days post-first dose administration [21]. Another study collected SAEs for 180 days following the administration of the first dose [27].

Most trials collected AE data using diary cards provided to the guardians with instructions on how to collect solicited and unsolicited events [16,17,19,21,22,23,25,26,28,29,30], sometimes supplemented by memory aids [33]. One study employed home visits by trained field workers to collect AE data [32]. Studies differed in which solicited and unsolicited outcomes they requested during the follow-up period. The most common local solicited reactions requested by investigators included tenderness or pain, erythema, and swelling. Systemic solicited reactions requested by investigators often included fever; drowsiness in children 6–35 months old; and headache in children 3–8 years old. Safety outcomes were reported either pooled across both doses [14,15,16,20,21,24,27,29,31] or separated by first and second dose [17,22,23,25,26,28,30,32,33].

Several studies provided thermometers for temperature monitoring [30,31]. One study pre-determined the time of day at which the temperature should be recorded; however, it was not reported if a thermometer was provided for caregivers [28]. Fever definitions were heterogeneous: ≥38.0 °C [20,21,23,24,25,26,29], ≥37.8 °C [33], ≥37.5 °C [17,19,22,28,30,31,32], ≥37.3 °C [14], and ≥37.1 °C [15,16,18].

### 3.5. Reported Safety Outcomes

Systemic reactogenicity outcomes are presented as reported by each trial for 6–35 months old, and for children 3–8 years old in Table 3. For Table 3, the data is presented as the percentage of participants (%) who experienced the event in that category with, in parentheses, participants who experienced one event (n) over the total participants (N) in that study. It was noted whether outcomes were pooled or reported stratified by dose number (first vs. second dose). Because AE definitions, solicited windows, and dose handling varied substantially across trials, no cross-trial synthesis was conducted. Several trials pooled systemic events across both doses or across multiple vaccine arms, whereas others reported events separated by dose number, HA content, or valency. The heterogeneity in reporting of pooling precluded a consistent dose or valency synthesis of systemic reactogenicity.

Across both age bands, reactions at the injection site (tenderness or pain, erythema, and induration or swelling) were the most frequently reported AEs. These AEs were usually solicited during days 0–7 after each vaccination. In younger children (6–35 months), systemic events such as irritability, drowsiness, and decreased appetite were commonly monitored, whereas in older children (3–8 years), headache, myalgia, and malaise were more frequently reported. However, trials differed in which systemic events were solicited, whether these were stratified by dose number, and whether results were pooled across vaccine formulations, limiting direct comparison of AE patterns by age, dose, or vaccine platform.

No vaccine-related SAEs were reported in any children 3–8 years old. A total of four vaccine-related SAEs were reported in children aged 6–35 months old; all trials resolved without participant withdrawal. Of these, one study had two reported vaccine-related SAEs in the quadrivalent trials, with one child developing pneumonia and another an upper respiratory tract infection (URTI) three days after the second vaccination [15]. This study considered the SAEs to be vaccine-related due to the temporality of the events; however, the etiology was reported as unknown by the investigators [15]. Another study with a quadrivalent influenza vaccine (QIV) had one reported SAE, which was a partial seizure associated with fever six hours after the first dose; it resolved, and that same child received the second dose twenty-eight days later and reported no SAEs [23]. Lastly, another study had one participant who experienced a QIV (a benign febrile convulsion due to a URTI). The child made a full recovery and was not withdrawn from the study [24].

### 3.6. Immunogenicity Data Collection

Immunogenicity data collection varied across studies. Limited stratification by influenza vaccine-primed status restricted Section 3.6 and Section 3.7 to thirteen studies overall. Limited age-stratified immunogenicity reporting in children 3–8 years old restricted Section 3.6 and Section 3.7 to six studies.

For studies in children 6–35 months old, immunogenicity data were consistently collected at baseline (pre-dose 1) to establish pre-existing hemagglutination inhibition assays (HAI) titers. Immunogenicity data collection after first dose was carried out on the following days: day 28 [32], day 42 [28], day 50 [26], day 56 [15,16,17,22,25,32,33], day 180 [28], and day 201 [28]. All studies for children only 3–8 years old collected baseline immunogenicity data before the first dose. For studies in children only 3–8 years old, immunogenicity data collection post first dose occurred on the following days: day 28 [29], day 56 [18,29], and day 58 [14]. In mixed-age cohorts (6 months to 8 years), all studies sampled baseline immunogenicity data before first vaccination and on day 56 after the first dose [20,21,27,30,31].

Across studies, HAI assays were used to establish geometric mean antibody titers (GMTs) [15,16,17,20,21,22,25,26,27,28,29,30,31,32,33], with validation reported from the Centers for Disease Control and Prevention (CDC): Quality Assurance Division [30,31,33], Cliantha Research [22], or the National Institute for Food and Drug Control [14]. Investigators also utilized geometric mean titer ratios (GMTRs), defined as the individual ratio of post- to pre-vaccination GMTRs, the post-vaccination time point varied across studies [16,20,21,26,27,28,29]. Some reports compared GMTs between experimental or control vaccine groups and defined this as GMTRs [14,15,17]. One study presented the geometric mean fold rise from pre- to post-vaccination [28]. Seroprotection rate (SPR) was the proportion of participants with an HAI ≥ 1:40 [14,15,16,17,21,22,25,28,29,30,31,32,33]. The seroconversion rate (SCR) was the proportion of participants with a pre-vaccination titer of <1:10 and a post-vaccination titer of more than or equal to ≥1:40, or a pre-vaccination titer of ≥1:10 and a ≥4-fold increase in post-vaccination titer [14,15,16,17,20,21,22,25,26,27,28,29,30,31,32,33]. A subset of studies defined seropositivity as an HAI of ≥1:10 [25,28,31,32].

### 3.7. Immunogenicity Outcomes

Immunogenicity outcomes are presented exactly as reported by each trial for children 6–35 months old and for children 3–8 years old in Table 4. Reported metrics include GMTs, GMTRs, GMFRs, SPR, and SCR. It was noted whether outcomes were stratified by age group, priming status, dose number, HA content, valency, and vaccine platform, or pooled across these categories. Because sampling schedules, priming status strata, and reporting frameworks varied, a formal cross-trial synthesis was not conducted.

Across studies, vaccination consistently increased HAI GMTs from baseline in both age groups for A(H1N1), A(H3N2), and B lineages. Most trials reported post-vaccination SPR and SCR that were consistent with licensure expectations for the formulations being tested at their designated peak time point (typically day 28, 42, or 56/58), but the response extent varied by strain and study. In younger children aged 6–35 months, trials often enrolled mixed cohorts of influenza vaccine-naïve and primed participants and sometimes pooled immunogenicity outcomes across priming strata, limiting interpretation of dose–response relationships.

When immunogenicity data were stratified by age group, children aged 3–8 years old generally had higher baseline titers and maintained high post-vaccination GMTs, SPR, and SCR across strains, indicating prior exposure in participants. In contrast, children aged 6–35 months old typically showed lower baseline titers and a larger relative fold rise after vaccination. The number of studies providing age-stratified GMTs, SPR, and SCR estimates by vaccine type or dose was limited, and statistical comparisons between age strata were not frequently reported. Thus, age-specific comparative conclusions could not be made.

Only a handful of trials directly compared different HA doses or valencies within the same studies. In those studies, immunogenicity outcomes were sometimes reported pooled across doses. In studies where higher-dose formulations were evaluated, post-vaccination GMTs and SCR were often numerically higher in the 15 µg arms, but inconsistencies in the reporting of variability limited the ability to determine whether these differences were clinically meaningful. Similarly, only a small number of studies compared trivalent versus quadrivalent inactivated vaccines or split-virion versus subunit formulations.

Immunogenicity reporting by vaccine type and adjuvant status was also incomplete. A few trials evaluated adjuvanted formulations in specific age bands, often alongside non-adjuvanted comparators. In these studies, higher GMTs were achieved in adjuvanted formulations versus non-adjuvanted in at least one influenza strain. However, outcomes were not always stratified by priming status or dose number, and statistical comparisons were inconsistently presented. Most trials focused on egg-based, non-adjuvanted split-virion quadrivalent or trivalent vaccines and did not allow for systematic comparisons of adjuvanted versus non-adjuvanted vaccines in ages 6–35 months old and 3–8 years old.

In summation, these findings indicate that seasonal, egg-based inactivated influenza vaccines elicit measurable HAI responses and generally achieve high SPR and SCR in healthy children aged 6 months to 8 years old. However, across all studies, the way immunogenicity data is defined, stratified, and reported across trials is highly heterogeneous. Limited age-stratified reporting; frequent pooling of results across doses, valencies, priming strata; and inconsistent use of standardized definitions and time points substantially constrained the ability to compare immunogenicity profiles by age group, dose, valency of vaccine platforms. These patterns directly address our objective to map how safety and immunogenicity outcomes are defined, collected, stratified, and reported in pediatric influenza vaccine trials and identify reporting gaps that limit cross-trial comparability.

### 3.8. Risk of Bias

Risk of bias in randomized trials (RoB 2) assessments were predominantly low-risk across randomized trials, with some concerns mainly stemming from the randomization process and selection of the reported results. Occasional concerns due to missing outcome data were noted as shown in Appendix A; risk from deviation from intended interventions and measurement of outcomes were generally low.

Risk of bias in non-randomized studies (ROBINS-I V2) assessments indicated a predominantly serious overall risk of bias among non-randomized studies, driven predominantly by confounding and selection of reported results. Risks due to missing data were generally low, while other domains ranged from low to moderate, as shown in Appendix A.

## 4. Discussion

This descriptive systematic review mapped how safety and immunogenicity outcomes are defined, collected, stratified, and reported in clinical trials of seasonal, egg-based inactivated influenza vaccines in healthy children 6 months to 8 years of age (Table 3 and Table 4). The 20 evaluated clinical trials confirmed that inactivated, intramuscular seasonal influenza vaccines have a favorable safety and immunogenicity profile in children aged 6 months to 8 years old. Substantial heterogeneity in outcome definitions, follow-up windows, and dosing strata constrains the ability to answer finer-grained questions by age, dose, vaccine valency, and formulation. These reporting differences constrained our capacity to perform meta-analysis and to critically address in a statistically robust way how safety and immunogenicity vary by age band, vaccine type, antigen content, and adjuvanted use. Thus, the central contribution of this review is not only to reaffirm the generally well-tolerated seasonal pediatric influenza vaccines but to identify concrete reporting gaps that restrict cross-trial comparability for evidence synthesis.

When examined descriptively, safety outcomes were reassuring across both age bands. In children 3–8 years old, pain at the injection site and headache were among the more frequently reported SE. No vaccine-related SAEs were reported in this age group. In children 6–35 months old, local reactions were commonly solicited in the first 7 days, while systemic reactions varied by study and age band. Only four vaccine-related SAEs were reported in children 6–35 months old, all of which resolved without permanent participant withdrawal. Within this overall favorable safety profile, we did not observe a consistent pattern of increased SAEs with higher antigen doses, quadrivalent versus trivalent formulations, or the use of MF59 adjuvant in vaccine-naïve children.

Trials consistently recognized the need for age-appropriate dosing, and all included vaccines used 7.5 µg or 15 µg of hemagglutinin per strain. Relatively few studies fully stratified solicited and unsolicited adverse events by dose within the same vaccine platform, and even fewer reported safety by both dose and age band. Almost half of the trials in children 6–35 months old and more than half of the trials in children 3–8 years old did not collect solicited reactions separately after each dose.

In studies that reported both doses separately, mild local reactions sometimes appeared more frequently after the first dose, with smaller increments or stable rates after the second dose, but these patterns were not consistently analyzed or highlighted by the original trial authors. The lack of standardized reporting by dose number and by age band makes it difficult to answer practical questions that are highly relevant to clinicians and parents, such as whether the second priming dose is systematically more reactogenic in very young children or whether certain formulations tend to produce more systemic reactions only after repeat exposure. While the data does not raise any new safety concerns for any evaluated formulation, it does not allow us to confidently qualify differential safety profiles by age, dose, or vaccine type.

Across the 20 trials, immunogenicity was typically assessed using HAI assays, with outcomes reported as SPR, SCR, GMT, and GMTR before and after vaccination. Most studies reported meeting commonly applied regulatory criteria for pediatric populations for at least two of the three vaccine strains evaluated, particularly in older children. Methodological choices also affected safety interpretation. Some trials presented strain-specific SPR and SCR, while others reported only SCR or geometric mean fold increase, and sampling time points ranged from 21 to 56 days post-vaccination. A small number of trials reported longer-term persistence of antibody responses; most did not.

Within individual trials, higher antigen doses or adjuvanted formulations often produced numerically higher SPR or SCR, especially for A(H1N1) and A(H3N2) strains, but the magnitude of these differences varied. MF59-adjuvanted trivalent vaccines in vaccine-naïve toddlers in rural Senegal also showed higher antibody responses than non-adjuvanted full-dose comparators, particularly for B strains, again with acceptable safety profiles [16,32]. In children 3–8 years old, quadrivalent inactivated vaccines were non-inferior to trivalent comparators for shared A and B strains, and they provided broader coverage for the additional B lineage [21]. Few studies stratified immunogenicity outcomes by prior vaccination status, even when priming status was recorded at baseline. As a result, it is difficult to determine whether the higher immunogenicity observed in older age groups or in certain formulations reflects antigen content, adjuvant effects, prior exposure, or a combination of these factors.

Trials by Hu et al., Langley et al., and Wang et al. evaluated co-reactivity responses across Yamagata and Victoria B-lineages in trivalent formulations by reporting both B lineages HI GMTs, including the lineage not contained in the respective TIV [15,17,23]. Within these studies, GMTs rose substantially for the respective vaccine B lineage and showed measurable increases against the alternate B lineage, providing rare strain-specific insight into cross-lineage responses in children. These studies illustrate a practical standard for trivalent immunogenicity assessment: GMTs, SCRs, and SPRs should be reported for all circulating B lineages and not only the one included in the vaccine. This would enable the cross-study comparison of cross-reactivity, and the potential mismatch can be evaluated in pediatric populations.

The available immunogenicity data support the conclusion that seasonal inactivated vaccines induce robust antibody responses in healthy children aged 6 months to 8 years, but the data do not allow us to precisely quantify how immunogenicity varies by dose, valency, or adjuvant when age and priming status are taken into account. This limitation directly reflects the heterogeneity in how immunogenicity outcomes were defined, collected, and stratified, and it underscores the need for more standardized pediatric immunogenicity reporting in future trials.

Our mapping shows that almost all trials collected solicited local and systemic reactions for at least seven days after each dose, but follow-up windows for unsolicited adverse events ranged from 21 to 30 days, and serious adverse events were followed from 56 to 180 days. A few studies classified systemic AEs recorded within seven days of vaccine administration as vaccine-related based on timing alone, without a formal causality assessment. This approach can misrepresent risk. In contrast, several studies in this review adjudicated the causality for systemic events from the outset, while others limited causality assessments to unsolicited events only. Fever thresholds ranged from 37.1 °C to 38.0 °C, and the specific systemic symptoms included in solicited profiles varied by study and age band. Some trials provided thermometers and standardized instructions for temperature measurement, while others relied on caregiver recall or did not report measurement methods. Reporting strategies also differed: several studies reported “any local” or “any systemic” event without disaggregating individual symptoms, whereas others reported only a subset of key symptoms.

These differences in definitions and follow-up windows matter because they directly affect reported event frequencies and the apparent safety profile of a vaccine. A formulation evaluated with a lower fever threshold and longer unsolicited follow-up period may appear “less safe” than one evaluated under narrower and less sensitive criteria, even if the underlying biological reactogenicity is similar. Without harmonization of outcome definitions and observation periods, cross-trial comparisons by vaccine type, valency, or dose are inherently confounded by methodological differences. These findings therefore support the need for consensus pediatric safety endpoints that specify core solicited local and systemic events, minimum follow-up durations for each outcome type, and standardized fever thresholds.

A similar pattern emerged for immunogenicity. Few trials explicitly referenced regulatory criteria when reporting their outcomes, and even when they did, the selected endpoints and time points were not uniform. Assays, laboratories, and cut-offs differed, and most studies did not describe assay validation in detail. Almost no trials stratified immunogenicity simultaneously by age band, dose, and priming status, even when those factors were measured. This lack of standardized, multi-strata immunogenicity reporting makes it difficult to answer targeted questions, such as whether 7.5 µg versus 15 µg HA per strain is sufficient for seroprotection in vaccine-naïve 6–35 month old or whether quadrivalent formulations confer any incremental immunogenic benefit compared with trivalent formulations in older children.

Our review shows that the main barrier to answering these clinically important questions is not simply the number of available trials but how outcomes are operationalized and reported within them. In this sense, our findings help explain why, despite decades of pediatric influenza vaccination, key comparative questions remain unresolved for frontline immunoprophylaxis specialists.

The evidence gaps identified in this review point to several concrete improvements for future clinical trials. First, standardized safety outcome definitions should be adopted, including a core set of solicited local and systemic events, harmonized fever thresholds, and minimum follow-up windows for solicited, unsolicited, and serious adverse events. Second, all pediatric trials should pre-specify stratified analyses by age band (e.g., 6–35 months versus 3–8 years), antigen dose, valency, and priming status, and they should present these results in a disaggregated format that allows cross-trial comparison. Third, immunogenicity assessments should use clearly described and, when possible, standardized assays and cut-offs, with consistent post-vaccination sampling time points and explicit reporting of seroconversion, seroprotection, and geometric mean titers for each strain. Finally, trials that test adjuvanted formulations or preservatives such as AS03, MF59, or thiomersal should systematically report both safety and immunogenicity outcomes by formulation to allow a balanced assessment of benefit versus reactogenicity, especially in the youngest age bands.

Adopting these practices would not only strengthen individual trials but would also enable more robust evidence synthesis in future systematic reviews and meta-analyses. In turn, this would better support the development of age-appropriate dosing recommendations, the evaluation of new formulations, and the identification of specific subgroups of children who might benefit from tailored vaccination strategies.

Future pediatric influenza vaccine trials should opt to apply a pre-specified causality algorithm or checklist, such as the causality assessment checklist recommended by the WHO [34]. Future studies should prespecify harmonized solicited lists, standardize fever thresholds, and stratify results by dose number and priming status. These steps would help improve comparability across studies and the utility of pediatric safety and immunogenicity evidence for policy decision making.

### Strengths and Limitations

This review maps both safety and immunogenicity outcomes across pediatric inactivated influenza vaccines with transparent, study-level reporting. We followed JBI and PRISMA 2020 methods, used dual screening, applied standardized extraction, and separated results by age band and dose when the published data allowed [13]. We reproduced each study’s denominators, follow-up windows, and assay validation sources, and, where possible, we converted percentages and graphs into counts for clarity. Risk of bias was systematically appraised using RoB 2 for randomized trials and ROBINS-I (V2) for non-randomized studies. Rather than forcing pooled estimates across heterogeneous outcome definitions and time points, we explicitly flagged differences in sampling days, priming status, and pooling of doses. The tables and Appendix A provide granular detail that can be reused by others, and they highlight specific targets for harmonization in future trial protocols.

This review is limited by features inherent to the underlying evidence base. Heterogeneity in outcome definitions and follow-up windows across trials; frequent pooling of outcomes across doses; incomplete reporting of vaccination primed status; and variable immunogenicity schedules all restrict the extent to which results can be compared or synthesized across studies. Fever thresholds and causality adjudication methods were inconsistently reported or incompletely described in several trials. All included studies enrolled healthy children, and high-risk pediatric populations were rarely represented in trial designs, which limits generalizability to children with chronic conditions or immunocompromising states. Several trials received industry funding, and in some cases, sponsors were involved in data analysis and reporting. These factors may influence outcome selection, analytic choices, and reporting, and they should be considered when interpreting our findings.

## 5. Conclusions

While inactivated, intramuscular seasonal influenza vaccines demonstrated a favorable safety profile in healthy children 6 months to 8 years old across the 20 included trials. However, the clinical trial literature remains insufficiently harmonized to support robust cross-study comparisons of safety and immunogenicity by age, dose, formulation, or valency. Our primary contribution shows that heterogeneous reporting frameworks, limited dose-specific and priming-specific data, and variable causality assessments and immunogenicity schedules substantially limit cross-trial comparability. These reporting gaps make it difficult to answer detailed questions about how safety and immunogenicity vary by age band, vaccine valency, antigen content, and dosing schedule, despite the existence of multiple trials.

Future research should extend follow-up windows when feasible, include high-risk populations, and adopt harmonized outcome definitions and adjudication procedures. Trials should be designed and reported in ways that preserve key stratifications by age, priming status, dose, and valency and that use standardized safety and immunogenicity endpoints. By aligning trial methods and reporting conventions, the field can move beyond confirming that pediatric influenza vaccines are safe and immunogenic to identifying the most effective and acceptable vaccination strategies for children in different clinical and epidemiologic contexts.

## Figures and Tables

**Figure 1 vaccines-14-00032-f001:**
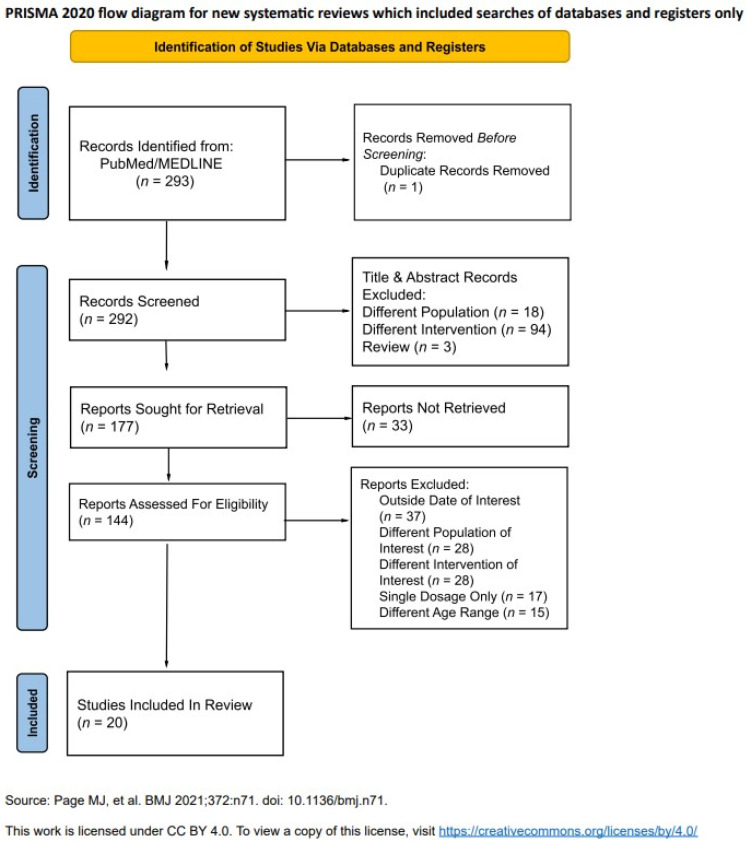
PRISMA 2020 flow diagram for new systematic reviews, which included searches of databases and registers only. Diagram showing the study selection flow from initial search to final inclusion [13]. Source: Page MJ, et al. BMJ 2021; 372: n71. doi: 10.1136/bmj.n71. This work is licensed under CC BY 4.0. To view a copy of this license, visit https://creativecommons.org/licenses/by/4.0/ (accessed 9 July 2025).

**Figure 2 vaccines-14-00032-f002:**
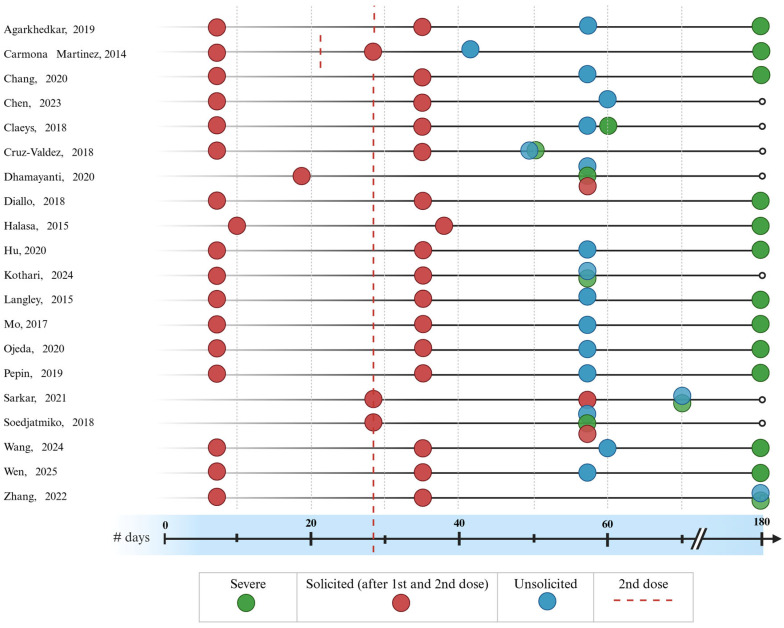
Duration of safety surveillance and follow-up periods across included studies (*N* = 20). This duration dot plot illustrates the variability and consistency in safety monitoring intervals for the analyzed studies. Each horizontal line corresponds to an individual study, with specific markers indicating the completion of follow-up for different safety outcomes. Legend and symbols: red dots represent the end of the follow-up period for solicited adverse events (typically 7–14 days post-vaccination). Blue dots represent the end of the follow-up period for unsolicited adverse events. Green dots represent the end of the follow-up period for serious adverse events (SAEs). Red dotted vertical line indicates the timing of the second dose administration, providing context for the reactogenicity monitoring periods. Markers have been rendered with partial transparency to manage overlapping data points, indicating concurrent endpoints. Data concentration: the alignment of green marker at the far right of the scale reflects a standardized 6-month (180-day) long-term safety monitoring period across the majority (65%-13/20) of the evidence base [14,15,16,17,18,19,20,21,22,23,24,25,26,27,28,29,30,31,32,33].

**Table 3 vaccines-14-00032-t003:** Safety and reactogenicity outcomes of pediatric seasonal influenza vaccination, stratified by age and event type (n = 17 studies *).

Reference	Age Bands	Vaccine	Dose Number	EA Local % (n/N)				Systemic Event % (n/N)						Any Solicited Event
				Tenderness/Pain	Erythema	Swelling	Any Local Event	Fever	Vomiting/Nausea	Headache	Drowsiness	Loss of Appetite	Any Systemic Event	
Agarkhedkar, 2019 [20]	6–35 MO	Subunit 4V (7.5 µg HA)	Pooled	12 (12/100)	1 (1/100)	2 (2/100)	13 (13/100)	9 (9/100)	2 (2/100)	0 (0/29)	5.8 (1/71)	4.2 (3/71)	15 (15/100)	24 24/100
	3–8 YO	Subunit 4V (15 µg HA)	Pooled	40 (40/100)	3 (3/100)	3 (3/100)	41 (41/100)	8 (8/100)	-	4 (4/100)	-	-	17 (17/100)	48 (48/100)
Carmona Martinez, 2014 [28]	6–35 MO	Split-virion 3V-Vic (7.5 µg HA), AS03	1st	30 (6/20)	10 (2/20)	5 (1/20)	-	35 (7/20)	25 (5/20)	-	20 (4/20)	45 (9/20)	-	-
			2nd	30 (6/20)	30 (6/20)	25 (5/20)	-	45 (9/20)	15 (3/20)	-	20 (4/20)	25 (5/20)	-	-
Chang, 2020 [29]	3–8 YO	Split-virion 4V (15 µg HA)	Pooled	51.4 (57/111)	8.1 (9/111)	15.3 (17/111)	36.9 (41/111)	6.3 (7/111)	-	-	-	-	44.1 (49/111)	-
Chen, 2023 [14]	3–8 YO	Split-virion 4V (15 µg HA)	1st	8 (43/538)	2 (11/538)	3 (16/538)	7.8 (42/538)	5 (27/538)	3 (16/538)	4 (22/538)	-	-	8.9 (48/538)	14.7 (79/538)
			2nd				3.8 (18/538)						2.3 (11/538)	5.5 (26/538)
Claeys, 2018 [25]	6–35 MO	Split-virion 4V (15 µg HA) †	1st	15 (69/462)	19 (88/462)	7 (32/462)	-	18 (83/462)	-	-	19 (88/462)	20 (92/463)	-	-
			2nd	10 (42/420)	14 (59/420)	7 (29/420)	-	10 (42/420)	-	-	15 (63/420)	15 (63/420)	-	-
		Split-virion 4V (15 µg HA)	1st	17 (80/470)	18 (85/470)	9 (42/470)	-	19 (90/470)	-	-	16 (75/470)	15 (71/470)	-	-
			2nd	10 (44/421)	15 (63/421)	6 (25/421)	-	10 (44/421)	-	-	14 (56/421)	16 (67/421)	-	-
Cruz-Valdez, 2018 [26]	6–35 MO	Subunit 3V-Yam (15 µg HA), MF59	Pooled	-	-	-	53.3 (48/90)	-	-	-	-	-	55.6 (50/90)	68.9 (62/90)
		Split-virion 3V-Yam (15 µg HA)	Pooled	-	-	-	35.2 (32/91)	-	-	-	-	-	42.9 (39/91)	55.6 (50/91)
Diallo, 2018 [32]	6–35 MO	Subunit 3V-Yam (7.5 µg HA), MF59	1st	41 (32/78)	-	-	-	25.6 (20/78)	-	-	-	-	-	-
			2nd	16.4 (12/73)	-	-	-	30.1 (22/73)	-	-	-	-	-	-
		Split-virion 3V-Yam (15 µg HA)	1st	35.4 (28/79)	-	-	-	11.4 (9/79)	-	-	-	-	-	-
			2nd	19.1 (13/68)	-	-	-	13.2 (9/68)	-	-	-	-	-	-
Hu, 2020 [15]	6–35 MO	Subunit 4V (7.5 µg HA)	Pooled	0.09 (1/1160)	1.38 (16/1160)	0.69 (8/1160)	1.9 (22/1160)	29.31 (340/1160)	1.21 (14/1160)	-	-	1.47 (17/1160)	30.69 (356/1160)	31.98 (371/1160)
		Subunit 3V-Vic (7.5 µg HA)	Pooled	0.17 (1/580)	1.03 (6/580)	0.34 (2/580)	1.55 (9/580)	30 (174/580)	0.52 (3/580)	-	-	1.03 (6/580)	31.21 (181/580)	32.24 (187/580)
		Subunit 3V-Yam (7.5 µg HA)	Pooled	0 (0/580)	1.21 (7/580)	0 (0/580)	1.38 (8/580)	34.14 (198/580)	0.69 (4/580)	-	-	1.72 (10/580)	34.14 (198/34.14)	32.24 (187/580)
Kothari, 2024 [22]	6–35 MO	Split-virion 4V ‡ (15 µg HA)	1st	3.4 (6/174)	2.9 (5/174)	1.1 (2/174)	-	7.5 (13/174)	-	-	-	-	-	-
			2nd	1.7 (3/172)	2.3 (4/172)	0.6 (1/172)	-	3.5 (6/172)	-	-	-	-	-	-
		Split-virion 4V (15 µg HA)	1st	4.1 (7/172)	5.2 (9/172)	4.7 (8/172)	-	8.7 (15/172)	-	-	-	-	-	-
			2nd	2.3 (4/172)	0.6 (1/172)	1.2 (2/172)	-	2.3 (4/172)	-	-	-	-	-	-
Langley, 2015 [23]	6–35 MO	Subunit 4V (15 µg HA)	1st	25 (71/284)	2 (6/284)	1 (3/284)	-	14 (40/284)	-	-	25 (71/284)	27 (77/284)	-	-
			2nd	21 (60/284)	3 (9/284)	3 (9/284)	-	10 (28/284)	-	-	17 (48/284)	18 (51/284)	-	-
		Subunit 3V (15 µg HA)	1st	22 (63/287)	1 (3/287)	1 (3/287)	-	15 (43/287)	-	-	22 (63/287)	24 (69/287)	-	-
			2nd	22 (63/287)	3 (9/287)	3 (9/287)	-	9 (26/287)	-	-	17 (46/287)	20 (57/287)	-	-
Mo, 2017 [16]	6–35 MO	Split-virion 3V-Yam (7.5 µg HA)	Pooled	20 (30/150)	2 (3/150)	3 (5/150)	23.6 (33/140)	30 (45/150)	15 (23/150)	2 (3/150)	17 (26/150)	23 (35/150)	42.9 (60/140)	49.6 (70/141)
Ojeda, 2020 [27]	6–35 MO	Split-virion 4V (15 µg HA), Thio	Pooled	-	-	-	54 (27/50)	-	-	-	-	-	66 (33/50)	72 (36/50)
		Split-virion 4V (15 µg HA)	Pooled	-	-	-	48.1 (26/54)	-	-	-	-	-	46.3 (25/54)	59.3 (32/54)
	3–8 YO	Split-virion 4V (15 µg HA), Thio	Pooled	-	-	-	57.1 (16/28)	-	-	-	-	-	46.7 (14/30)	67.9 (19/28)
		Split-virion 4V (15 µg HA)	Pooled	-	-	-	59.3 (16/27)	-	-	-	-	-	55.6 (15/27)	70.4 (19/27)
Pepin, 2019 [24]	6–35 MO	Split-virion 4V (15 µg HA)	Pooled	-	-	-	39.9 (635/1591)	-	-	-	-	-	48.5 (772/1592)	-
Sarkar, 2021 [21]	6–35 MO	Split-virion 4V (15 µg HA)	Pooled	6.9 (7/102)	2 (2/102)	2.9 (3/102)	-	17.5 (18/102)	-	2 (2/102)	-	-	-	-
		Split-virion 3V-Vic (15 µg HA)		3.8 (4/104)	0 (0/104)	0 (0/104)	-	8.7 (8/104)	-	0 (0/104)	-	-	-	-
	3–8 YO	Split-virion 4V (15 µg HA)	Pooled	10 (10/97)	2 (2/97)	0 (0/97)	-	11.3 (11/97)	-	4.1 (4/97)	-	-	-	-
		Split-virion 3V-Vic (15 µg HA)		16.7 (17/102)	7.8 (8/102)	2 (2/102)	-	13.7 (14/102)	-	3.9 (4/102)	-	-	-	-
Wang, 2024 [17]	6–35 MO	Split-virion 4V (7.5 µg HA)	1st	0 (0/659)	4.86 (32/659)	0 (0/659)	-	7.28 (48/659)	1.37 (9/659)	-	-	0.91 (6/659)	-	-
			2nd	0.31 (2/659)	5.95 (38/659)	0.16 (1/659)	-	6.26 (40/659)	1.56 (10/659)	-	-	0.31 (2/659)	-	-
		Split-virion 4V (15 µg HA)	1st	0.15 (1/660)	6.06 (40/660)	0.3 (2/660)	-	6.67 (44/660)	1.36 (9/660)	-	-	0.61 (4/660)	-	-
			2nd	0 (0/660)	6.24 (40/326)	0.16 (1/660)	-	8.11 (52/660)	1.40 (9/660)	-	-	0 (0/660)	-	-
		Split-virion 3V-Vic (7.5 µg HA)	1st	0.31 (1/326)	3.68 (12/326)	0 (0/326)	-	8.59 (28/326)	1.84 (6/326)	-	-	1.23 (4/326)	-	-
			2nd	0 (0/326)	5.48 (17/326)	0 (0/326)	-	7.10 (22/326)	1.61 (5/326)	-	-	0.65 (2/326)	-	-
		Split-virion 3V-Yam (15 µg HA)	1st	0.30 (1/329)	7.29 (24/329)	0.61 (2/329)	-	8.51 (28/329)	2.13 (7/329)	-	-	0.91 (3/329)	-	-
			2nd	0.32 (1/329)	6.39 (20/329)	0.64 (2/329)	-	6.71 (21/329)	0.96 (3/329)	-	-	0 (0/329)	-	-
Wen, 2025 [18]	3–8 YO	Split-virion 4V (15 µg HA)	1st	0.83 (1/120)	-	0 (0/120)	-	4.17 (5/120)	-	-	-	-	-	5 (6/120)
			2nd	0 (0/120)	-	0 (0/120)	-	0 (0/120)	-	-	-	-	-	0 (0/120)
Zhang, 2022 [19]	3–8 YO	Subunit 4V (15 µg HA)	1st	2.75 (11/400)	0.75 (3/400)	0.5 (2/400)	3.75 (15/400)	2.25 (9/400)	0.75 (3/400)	0.25 (1/400)	-	-	6.25 (25/400)	9.75 (39/400)
			2nd	3.83 (15/392)	0.26 (1/392)	1.28 (5/392)	5.61 (22/392)	0.51 (2/392)	0.26 (1/392)	0.26 (1/392)	-	-	2.55 (10/392)	8.16 (32/392)
		Split-virion 4V (15 µg HA)	1st	5.01 (20/399)	0.75 (3/399)	0.5 (2/399)	6.25 (25/399)	5.01 (20/399)	0 (0/399)	0.75 (3/399)	-	-	8.02 (32/399)	13.78 (55/399)
			2nd	2.03 (8/395)	-	-	2.03 (8/395)	2.03 (8/395)	0.51 (2/395)	-	-	-	4.05 (16/395)	6.08 (24/395)

This table summarizes the proportion of participants who experienced solicited adverse events (AEs) following vaccination, stratified by age group, vaccine formulation, and dose number. Adverse events are categorized as Local (at the injection site) or Systemic (general symptoms). Key definitions: vaccine identifies the full vaccine formulation, including type (split-virion or subunit), valency (3V or 4V), HA content in µg, and the presence of adjuvant (e.g., AS03 or MF59). Data Reporting: Numbers in the columns Tenderness/Pain through Loss of Appetite represent the percentage (%) and the number of participants (n) who experienced the event over the total number (N) of participants in that study. Summary columns (Any Local Event, Any Systemic Event, Any Solicited Event): These numbers represent the percentage of participants (%) who experienced at least one event in that category, with participants who experienced one event over the total participants reported for that group in parentheses. Pooled indicates that the results represent the combined data from multiple vaccine groups or multiple doses within the study. Abbreviations: MO, months old; YO, years old; HA, hemagglutinin; µg, microgram; AS03/MF59, specific adjuvants; 1st or 2nd, first or second dose; 3V-Vic, B/Victoria; 3V-Yam, B/Yamagata; (-), Data not reported. * Studies reported systemic reactogenicity outcomes as immediate, intermediate, and delayed; thus, it was not possible to determine if the same individual are accounted twice for more than one category (Dhamayanti, 2020 [30], Halasa, 2015 [33], Soedjatmiko, 2018) [31]. † (Claeys, 2018) study evaluated an approved method for split-virion vaccine formulations against a new experimental process [25]. ‡ (Kothari, 2024) trial compared two egg-based, split-virion quadrivalent inactivated vaccines (15 µg HA per strain), identical in formulation and dose and differing only by manufacturer [22].

**Table 4 vaccines-14-00032-t004:** Immunogenicity outcomes (GMT, seroconversion rate, and seroprotection rate) of seasonal influenza vaccines in children (*n* = 16 studies).

Reference	Age Bands	Vaccine	Subtype/Linage	HAI GMT				GMTR		SCR (%)		SPR (%)		
				D0	D28	D42	D56	D28	D56	D28	D56	D0	D28	D56
Agarkhedkar, 2019 [20]	6–35 MO	Subunit 4V (7.5 µg HA)	A(H1N1)	15	-	-	700	-	31	-	85	-	-	-
			A(H3N2)	40	-	-	1000	-	15	-	85	-	-	-
			B/Victoria	20	-	-	650	-	21	-	85	-	-	-
			B/Yamagata	10	-	-	950	-	52	-	92	-	-	-
	3–8 YO	Subunit 4V (15 µg HA)	A(H1N1)	70	-	-	2000	-	18	-	88	-	-	-
			A(H3N2)	400	-	-	3500	-	8	-	78	-	-	-
			B/Victoria	40	-	-	2500	-	42	-	90	-	-	-
			B/Yamagata	80	-	-	3000	-	38	-	94	-	-	-
Carmona Martinez, 2014 * [28]	6–35 MO	Split-virion 3V-Vic (7.5 µg HA), AS03	A(H1N1)	20	-	1000	-	-	65	-	100	30	-	100
			A(H3N2)	20	-	800	-	-	45	-	100	30	-	100
			B/Victoria	8	-	700	-	-	75	-	100	5	-	100
Chang, 2020 [29]	3–8 YO	Split-virion 4V (15 µg HA) Naive	A(H1N1)	33.7	167.2	-	358.4	5	10.6	56.4	90.9	54.4	65.5	96.4
			A(H3N2)	29.6	255.1	-	616.3	8.3	20.9	69.1	90.9	54.6	74.6	98.2
			B/Victoria	57.7	190.9	-	582.3	3.3	10.1	54.6	83.6	76.4	98.2	100
			B/Yamagata	129.1	266.6	-	422.2	2.1	3.3	32.7	49.1	100	98.2	100
		Split-virion 4V (15 µg HA) Primed	A(H1N1)	64	328	-	269.1	5.12	-	53.6	48.2	67.9	98.2	96.4
			A(H3N2)	101.2	601.6	-	586.9	5.94	-	64.3	62.5	71.4	96.4	98.2
			B/Victoria	29.7	185.6	-	210.1	6.25	-	71.4	78.6	41.1	94.6	98.2
			B/Yamagata	64.4	316.1	-	298.9	4.6	-	66.1	64.3	73.2	98.2	98.2
Chen, 2023 * [14]	3–8 YO	Split-virion 4V (15 µg HA)	A(H1N1)	-	-	-	365.38	-	-	-	87.7	-	-	95.22
			A(H3N2)	-	-	-	598.94	-	-	-	94.99	-	-	96.81
			B/Victoria	-	-	-	90.91	-	-	-	88.84	-	-	91.34
			B/Yamagata	-	-	-	152.36	-	-	-	88.61	-	-	93.17
Claeys, 2018 [25]	6–35 MO	Split-virion 4V (15 µg HA) †	A(H1N1)	11.1	-	-	97.5	-	-	-	66.6	19.5	-	70.1
			A(H3N2)	7.5	-	-	45.2	-	-	-	50.3	12.8	-	53.7
			B/Victoria	5.7	-	-	32.1	-	-	-	49.4	3.9	-	49.5
			B/Yamagata	8.3	-	-	100.8	-	-	-	73.8	12.3	-	76.2
		Split-virion 4V (15 µg HA)	A(H1N1)	11.2	-	-	105.3	-	-	-	55.8	19.6	-	67.7
			A(H3N2)	8.4	-	-	56.3	-	-	-	49.9	15.8	-	60.7
			B/Victoria	7.9	-	-	106.6	-	-	-	75.9	3.8	-	50.8
			B/Yamagata	5.7	-	-	37.7	-	-	-	321	11.6	-	77.5
Cruz-Valdez, 2018 * [26]	6–35 MO	Subunit 3V-Yam (15 µg HA), MF59	A(H1N1)								97.2			
			A(H3N2)								90.1			
			B/Yamagata								81.7			
		Split-virion 3V-Yam (15 µg HA)	A(H1N1)								77.1			
			A(H3N2)								81.4			
			B/Yamagata								17.1			
Dhamayanti, 2020 [30]	6–35 MO	Subunit 4V (15 µg HA)	A(H1N1)	25	-	-	400	-	-	-	96.8	22.5	-	97.5
			A(H3N2)	40	-	-	600	-	-	-	97.4	35.8	-	98.3
			B/Victoria	25	-	-	200	-	-	-	90.4	21.7	-	92.5
			B/Yamagata	20	-	-	90	-	-	-	84.3	4.2	-	85
	3–8 YO	Subunit 4V (15 µg HA)	A(H1N1)	75	-	-	850	-	-	-	10	70.9	-	100
			A(H3N2)	90	-	-	1050	-	-	-	100	82.1	-	100
			B/Victoria	35	-	-	400	-	-	-	96.7	31.3	-	97.8
			B/Yamagata	40	-	-	400	-	-	-	92	34.3	-	94.8
Diallo, 2018 [32]	6–35 MO	Subunit 3V-Yam (7.5 µg HA), MF59	A(H1N1)	-	-	-	-	-	-	78.7	100	6.6	78.7	100
			A(H3N2)	-	-	-	-	-	-	85.2	96.7	49.2	88.5	100
			B/Yamagata	-	-	-	-	-	-	54.1	90.2	27.9	63.9	100
		Split-virion 3V-Yam (15 µg HA)	A(H1N1)	-	-	-	-	-	-	10	85	3.3	10	85
			A(H3N2)	-	-	-	-	-	-	73.3	100	53.3	75	98.3
			B/Yamagata	-	-	-	-	-	-	28.3	70	35	41.7	90
Halasa, 2015 [33]	6–35 MO	Split-virion 3V-Vic (7.5 µg HA)	A(H1N1)	10.4	-	-	181.5	-	-	-	78	-	-	85
			A(H3N2)	10.5	-	-	12.9	-	-	-	7	-	-	15
			B/Victoria	12	-	-	27.4	-	-	-	31	-	-	44
		Split-virion 3V-Vic (15 µg HA)	A(H1N1)	8.8	-	-	187.2	-	-	-	85	-	-	89
			A(H3N2)	8.2	-	-	14.8	-	-	-	11	-	-	15
			B/Victoria	8.6	-	-	31.1	-	-	-	42	-	-	50
Hu, 2020 [15]	6–35 MO	Subunit 4V (7.5 µg HA)	A(H1N1)	20.03	-	-	329.9	-	-	-	87.46	39.66	-	92.05
			A(H3N2)	10.48	-	-	136.76	-	-	-	75.4	23.48	-	77.64
			B/Victoria	11.06	-	-	52.25	-	-	-	59.49	10.29	-	69.32
			B/Yamagata	17.9	-	-	104.3	-	-	-	71.84	37.42	-	85.03
		Subunit 3V-Vic (7.5 µg HA)	A(H1N1)	-	-	-	-	-	-	-	-	-	-	-
			A(H3N2)	-	-	-	-	-	-	-	-	-	-	-
			B/Victoria	11.49	-	-	61.02	-	-	-	66.85	10	-	78.15
			B/Yamagata	18.88	-	-	42.43	-	-	-	22.78	39.26	-	61.85
		Subunit 3V-Yam (7.5 µg HA)	A(H1N1)	-	-	-	-	-	-	-	-	-	-	-
			A(H3N2)	-	-	-	-	-	-	-	-	-	-	-
			B/Victoria	11.03	-	-	27.98	-	-		27.93	9.12	-	42.83
			B/Yamagata	20.29	-	-	126.66	-	-		75.42	39.66	-	88.27
Kothari, 2024 [22]	6–35 MO	Split-virion 4V (15 µg HA) ‡	A(H1N1)	-	-	-	905.1	-	-	-	100	-	-	100
			A(H3N2)	-	-	-	96.7	-	-	-	90.1	-	-	95.3
			B/Victoria	-	-	-	230	-	-	-	97.7	-	-	99.4
			B/Yamagata	-	-	-	217.3	-	-	-	97.7	-	-	99.4
		Split-virion 4V (15 µg HA)	A(H1N1)	-	-	-	719.8	-	-	-	100	-	-	100
			A(H3N2)	-	-	-	94.1	-	-	-	93	-	-	95.9
			B/Victoria	-	-	-	220.4	-	-	-	98.3	-	-	100
			B/Yamagata	-	-	-	227.7	-	-	-	91.8	-	-	97.1
Langley, 2015 [23]	6–35 MO	Subunit 4V (15 µg HA)	A(H1N1)	9.6			157.1				85.9	16.2		89.4
			A(H3N2)	17.4			159.4				72.2	32.7		81.3
			B/Victoria	10.6			111.4				73.9	19.7		76.1
			B/Yamagata	7.7			114.2				78.9	19.7		85.2
		Subunit 4V (15 µg HA)	A(H1N1)	9.8			61.2				53.7	16.4		58.9
			A(H3N2)	13.8			103				55.7	25.8		66.6
			B/Victoria	9.3			15.6				9.8	15.7		25.8
			B/Yamagata	7.2			107.2				77.4	8.4		79.8
Mo, 2017 [16]	6–35 MO	Split-virion 3V-Yam (7.5 µg HA)	A(H1N1)	12.8	-	-	152.2	-	11.6	-	86.4	31.3	-	88.8
			A(H3N2)	14.3	-	-	235.9	-	16.8	-	90.4	32	-	98.4
			B/Yamagata	5.9	-	-	108.5	-	18.5	-	93.5	1.3	-	93.5
Ojeda, 2020 ‡ [27]	6–35 MO	Split-virion 4V (15 µg HA), Thio	A(H1N1)	8	-	-	300	-	-	-	-	-	-	-
			A(H3N2)	15	-	-	500	-	-	-	-	-	-	-
			B/Victoria	8	-	-	250	-	-	-	-	-	-	-
			B/Yamagata	11	-	-	400	-	-	-	-	-	-	-
		Split-virion 4V (15 µg HA)	A(H1N1)	10	-	-	250	-	-	-	-	-	-	-
			A(H3N2)	10	-	-	300	-	-	-	-	-	-	-
			B/Victoria	7	-	-	200	-	-	-	-	-	-	-
			B/Yamagata	8	-	-	350	-	-	-	-	-	-	-
	3–8 YO	Split-virion 4V (15 µg HA), Thio	A(H1N1)	70	-	-	1000	-	-	-	-	-	-	-
			A(H3N2)	90	-	-	2000	-	-	-	-	-	-	-
			B/Victoria	20	-	-	700	-	-	-	-	-	-	-
			B/Yamagata	50	-	-	1050	-	-	-	-	-	-	-
		Split-virion 4V (15 µg HA)	A(H1N1)	30	-	-	600	-	-	-	-	-	-	-
			A(H3N2)	90	-	-	1000	-	-	-	-	-	-	-
			B/Victoria	20	-	-	500	-	-	-	-	-	-	-
			B/Yamagata	60	-	-	1025	-	-	-	-	-	-	-
Pepin, 2019 [24]	6–35 MO	Split-virion 4V (15 µg HA)	A(H1N1)	-	-	-	650	-	-	-	-	-	-	-
			A(H3N2)	-	-	-	1075	-	-	-	-	-	-	-
			B/Victoria	-	-	-	593	-	-	-	-	-	-	-
			B/Yamagata	-	-	-	997	-	-	-	-	-	-	-
Sarkar, 2021 [21]	6–35 MO	Split-virion 4V (15 µg HA)	A(H1N1)	-	-	-	766.4	-	-	-	100	-	-	100
			A(H3N2)	-	-	-	856.3	-	-	-	98	-	-	98
			B/Victoria	-	-	-	211.1	-	-	-	89.6	-	-	96
			B/Yamagata	-	-	-	72.6	-	-	-	83.3	-	-	90
		Split-virion 3V-Vic (15 µg HA)	A(H1N1)	-	-	-	814	-	-	-	91.8	-	-	100
			A(H3N2)	-	-	-	1111.2	-	-	-	95.9	-	-	100
			B/Victoria	-	-	-	113.9	-	-	-	87.8	-	-	85.1
			B/Yamagata	-	-	-	13.5	-	-	-	44.9	-	-	34.3
	3–8 YO	Split-virion 4V (15 µg HA)	A(H1N1)	-	-	-	854.3	-	-	-	93.8	-	-	100
			A(H3N2)	-	-	-	2031.9	-	-	-	89.6	-	-	100
			B/Victoria	-	-	-	169.5	-	-	-	89.6	-	-	91.7
			B/Yamagata	-	-	-	88.5	-	-	-	83.3	-	-	87.5
		Split-virion 3V-Vic (15 µg HA)	A(H1N1)	-	-	-	964.6	-	-	-	91.8	-	-	100
			A(H3N2)	-	-	-	2286.1	-	-	-	95.9	-	-	100
			B/Victoria	-	-	-	227.9	-	-	-	87.8	-	-	89.8
			B/Yamagata	-	-	-	30.1	-	-	-	44.9	-	-	51
Soedjatmiko, 2018 [31]	6–35 MO	Subunit 3V (15 µg HA), Thio	A(H1N1)	135	-	-	700	-	-	-	7.4	92.6	-	100
			A(H3N2)	175	-	-	790	-	-	-	2.2	97.8	-	100
			B/Yamagata	100	-	-	590	-	-	-	8.1	91.9	-	100
	3–8 YO	Subunit 3V (15 µg HA), Thio	A(H1N1)	180	-	-	1190	-	-	-	2.2	97.8	-	100
			A(H3N2)	190	-	-	590	-	-	-	2.2	97.8	-	100
			B/Yamagata	170	-	-	700	-	-	-	7.4	92.6	-	100
Wang, 2024 * [17]	6–35 MO	Split-virion 4V (7.5 µg HA)	A(H1N1)	11.76	-	-	185.35	-	-	-	89.14	-	-	92.76
			A(H3N2)	7.84	-	-	106.87	-	-	-	73.68	-	-	75.33
			B/Victoria	8.97	-	-	173.29	-	-	-	83.72	-	-	88.98
			B/Yamagata	8.56	-	-	85.86	-	-	-	75.49	-	-	79.28
		Split-virion 4V (15 µg HA)	A(H1N1)	13.71	-	-	226.15	-	-	-	226.15	-	-	94.18
			A(H3N2)	7.91	-	-	146.29	-	-	-	146.29	-	-	82.23
			B/Victoria	9.12	-	-	200.84	-	-	-	200.84	-	-	92.89
			B/Yamagata	8.48	-	-	104.08	-	-	-	104.08	-	-	85.95
		Split-virion 3V-Vic (7.5 µg HA)	A(H1N1)	13.57	-	-	215.13	-	-	-	215.13	-	-	93.71
			A(H3N2)	8.87	-	-	119.82	-	-	-	119.82	-	-	76.49
			B/Victoria	9.55	-	-	209.29	-	-	-	209.29	-	-	91.06
			B/Yamagata	9.23	-	-	43.35	-	-	-	43.35	-	-	58.28
		Split-virion 3V-Yam (7.5 µg HA)	A(H1N1)	14.97	-	-	212.76	-	-	-	212.76	-	-	93.09
			A(H3N2)	7.42	-	-	102.1	-	-	-	102.1	-	-	72.7
			B/Victoria	9.15	-	-	52.35	-	-	-	52.35	-	-	56.25
			B/Yamagata	8.7	-	-	94.27	-	-	-	94.27	-	-	78.95
Wen, 2025 [18]	3–8 YO	Split-virion 4V (15 µg HA)	A(H1N1)	20	400	-	800	-	-	82	100	40	85	98
			A(H3N2)	50	200	-	400	-	-	55	80	50	65	95
			B/Victoria	60	250	-	300	-	-	50	75	85	90	98
			B/Yamagata	20	200	-	200	-	-	60	90	25	75	90
Zhang, 2022 [19]	3–8 YO	Split-virion 4V (15 µg HA)	A(H1N1)				486.78				83.9			90.91
			A(H3N2)				700.28				87.53			94.03
			B/Victoria				36.95				69.61			70.39
			B/Yamagata				196.81				81.62			85.97
		Split-virion 4V (15 µg HA)	A(H1N1)				367.19				79.12			89.89
			A(H3N2)				504.17				87.89			93.30
			B/Victoria				31.26				68.04			69.85
			B/Yamagata				134.3				70.62			80.93

This table summarizes the main serological outcomes—including GMT, GMTR, seroconversion (SC), and seroprotection (SP) rates—achieved after vaccination. Results are stratified by the specific influenza strain/lineage and the dose number administered. Data is reported at multiple time points (D0, D28, D42, and peak response D56/58). Key definitions: subtype/linage indicates the specific influenza strain tested: A(H1N1) and A(H3N2) are Type A strains, and B/Victoria and B/Yamagata are Type B lineages. GMT is presented in the standard titer format (e.g., 905.2). It is the average antibody titer for a specific strain at a given time point. GMTR represents the fold increase in antibody titers from baseline (D0). Calculations typically use a threshold of ≥4×. Seroconversion is reported as a percentage (%) in the study. It signifies the proportion of participants achieving a specified increase in titer from baseline (usually ≥4×). Seroprotection is reported as a percentage (%) of protected participants (n). It signifies the proportion of participants achieving a protective antibody titer threshold (usually ≥1:40). Time points: D0, baseline (Day 0); D28 and D42, intermediate time points; D56/58, peak or late response time point. Abbreviations: MO, months old; YO, years old; HA, hemagglutinin; µg, microgram; GMT/GMTR, geometric mean titer/ratio; SCR, seroconversion rate; SPR, seroprotection rate; 3V-Vic, B/Victoria; 3V-Yam, B/Yamagata; (-), data not reported. * studies reported a late time point, as follows: Carmona, 2014: Day 42 [28], Chen, 2023: Day 58 [14], Cruz-Valdez, 2018: Day 49 instead of Day 56 [26]. † (Claeys, 2018) study evaluatesd an approved method for split-virion vaccine formulations against a new experimental process [25]. ‡ (Kothari, 2024) One trial compared two egg-based, split-virion quadrivalent inactivated vaccines (15 µg HA per strain), identical in formulation and dose and differing only by manufacturer [22].

## Data Availability

Data is contained within the article or Appendix A. The data extracted and analyzed in this systematic review are included in this article and its Appendix A. All source data are derived from previously published clinical trials cited in the References. Further inquiries can be directed to the corresponding author.

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
