# Peer review of "Mapping Pediatric Seasonal Influenza Vaccine Safety and Immunogenicity Evidence: A Systematic Review of Clinical Trials"

_vaccines, 2025, doi:10.3390/vaccines14010032_

Round 1
Reviewer 1 Report
Comments and Suggestions for Authors
This review conducted a systematic review to map clinical trial evidence regarding the safety and immunogenicity of influenza vaccines in healthy pediatric populations and offered some valuable suggestions for facilitating the comparison of future clinical trails.
- The assessment result for adverse events caused by influenza vaccines (Table S7 and S8) should be incorporated into the main text. Instead, Figure 2, Tables 1, 2, 3, 4, 5 are not required.
- In line 59, “of” should be replaced with “or”
- In figure 1, Records excluded:(n=115), the reason for records excluded was not indicated.
- In line 244, “ten” should be “nine”.
- In line 366, consistency of “6-35 months-old” should be maintained.
Reviewer 2 Report
Comments and Suggestions for Authors
This review article assesses the safety and immunogenicity of influenza vaccination in children aged 6 months to 8 years based on an analysis of publications up to 2025 using the JBI guidelines and PRISMA 2020 methodology.
Despite a meticulous and highly detailed analysis of the parameters studied from the clinical trial database included in this review, the authors conclude: "Seasonal, inactivated intramuscular influenza vaccines show a favorable safety and immunogenicity profile in healthy children aged 6 months to 8 years. Lack of standardized outcome definitions, dose-specific reporting, causality assessments, and naïve vs. primed specific reporting limits cross-trial comparability." In other words, the authors confirm the long- and previously known and already proven fact that all vaccines used in children are safe and provide protection against influenza, since a vaccine cannot be used in practice if it does not meet generally accepted criteria for safety and immunogenicity for different age groups.
Concerning data that the authors actually attempt to analyze in the "Results" section: no interesting and actual associations ware not reflected, only description even without statistics. Furthermore, the "Discussion" section also left unanswered questions regarding the safety and immunogenicity of influenza vaccines depending on the age of 6-35 months and 3-8 years, taking into account: valence (3- and 4-fold) of vaccines; vaccine type (split virion and subunit) vaccines; vaccines with different adjuvants (AS03, MF59, thiomersal); the amount of antigen in one vaccination dose (7.5 or 15 μg); differences in the incidence of local and systemic reactions after the first and second doses of the vaccine.
If the authors were unable to answer the questions they posed, which are extremely important for specialists involved in immunoprophylaxis, this means the adopted methodology requires improvement. I doubt that, despite the fact that influenza vaccination in children aged 6 months to 8 years has been used for decades, there are no answers to the question of its effectiveness depending on vaccine characteristics.
If the authors wish to address the current gaps in assessing the safety and immunogenicity of influenza vaccination in children aged 6 months to 8 years, then the title and objectives of the article should be changed, and the description of the results and discussion should emphasize the need to improve clinical trial protocols, specifically for analyzing the obtained results.
Reviewer 3 Report
Comments and Suggestions for Authors
The manuscript is generally well written; however, there is much content to be concerned about. I would like to comment as follows.
1) The manuscript does not clearly state the number of study participants included, the data sources used, nor the date up to which the literature search was conducted. These essential details should be clearly reported in the Abstract.
2) Several meta-analyses and systematic reviews on influenza vaccine safety have already been published. For example, Minozzi et al. (eClinicalMedicine, 2022; doi: 10.1016/j.eclinm.2022.101331), Carregaro et al. (BMC Infect Dis, 2023; doi: 10.1186/s12879-023-08541-0), and Wei et al. (Hum Vaccin Immunother, 2023; doi: 10.1080/21645515.2023.2256510) have evaluated immunogenicity and safety in various populations. The authors should summarize key findings from previous reviews and clearly highlight the knowledge gaps that their study aims to address, particularly in lines 78–79.
3) The stated aim of the study is “to map clinical trial evidence on influenza vaccine safety and immunogenicity in healthy pediatric populations aged 6 months to 8 years.” However, the results focus primarily on adverse events and do not adequately present or discuss immunogenicity outcomes. The authors should incorporate and evaluate immunogenicity data to align with the stated study objective.
4) The research question is not clearly articulated. It is unclear what intervention interest or whether the authors intend to compare safety outcomes across different age groups or vaccine dosage, or vaccine valency (the author mentioned in lines 80-81). The authors should clearly define the primary research question and specify the comparative framework in the results.
5) According to the aim of this study, the results should be described according to age group, vaccine dosage, and vaccine type. However, the author presented only adverse events according to age groups (6 months – 35 months and 3 – 8 years ), while results were summarized according to vaccine dose and vaccine type were lacking.
Round 2
Reviewer 2 Report
Comments and Suggestions for Authors
The authors took into account all the comments and made the necessary changes to the text. There are currently no questions or comments.
Author Response
We thank you for spending time to review the revision. We agree with you that the revision is acceptable for publication in this journal.